# Evaluation of systemic inflammatory response and lung injury induced by *Crotalus durissus cascavella* venom

Elen Azevedo[1,2]☯, Ricardo Gassmann Figueiredo[3], Roberto Vieira Pinto[4], Tarsila de Carvalho Freitas Ramos[5], Geraldo Pedral Sampaio[6], Rebeca Pereira Bulhosa Santos[6], Marcos Lázaro da Silva Guerreiro[1], Ilka Biondi[1]☯*, Soraya Castro Trindade[2,7]*

**1** Laboratory of Venomous Animals and Herpetology, Biology Department, State University of Feira de Santana, Feira de Santana, Brazil, **2** Postgraduate Program in Biotechnology, State University of Feira de Santana, Feira de Santana, Brazil, **3** Pulmonology Division, Department of Health, State University of Feira de Santana–UEFS, Feira de Santana, Brazil, **4** Pathological Anatomy Laboratory–LABSEAP, Cardiopulmonary Clinic, Novo Mundo, Brazil, **5** Laboratory of Oral Pathology, State University of Feira de Santana, Feira de Santana, Brazil, **6** Postgraduate Program in Immunology, Federal University of Bahia—UFBA, Salvador, Brazil, **7** Department of Health, Feira de Santana State University, Feira de Santana, Bahia, Brazil

☯ These authors contributed equally to this work.
* soraya.castrotrindade@gmail.com (SCT); ilkabiondi@gmail.com (IB)

**Data Availability Statement:** All relevant data are within the manuscript and its Supporting Information files.

## Abstract

This study investigated the systemic inflammatory response and mechanism of pulmonary lesions induced by *Crotalus durissus cascavella* venom in murine in the state of Bahia. In order to investigate T *helper* Th1, Th2 and Th17 lymphocyte profiles, we measured interleukin (IL) -2, IL-4, IL-6, IL-10, IL-17, tumor necrosis factor (TNF) and interferon gamma (IFN-γ) levels in the peritoneal fluid and macerated lungs of mice and histopathological alterations at the specific time windows of 1h, 3h, 6h, 12h, 24h and 48h after inoculation with *Crotalus durissus cascavella* venom. The data demonstrated an increase of acute-phase cytokines (IL-6 and TNF) in the first hours after inoculation, with a subsequent increase in IL-10 and IL-4, suggesting immune response modulation for the Th2 profile. The histopathological analysis showed significant morphological alterations, compatible with acute pulmonary lesions, with polymorphonuclear leukocyte (PMN) infiltration, intra-alveolar edema, congestion, hemorrhage and atelectasis. These findings advance our understanding of the dynamics of envenomation and contribute to improve clinical management and antiophidic therapy for individuals exposed to venom.

## Introduction

Accidents involving snakes have been considered a neglected disease by the World Health Organization (WHO) since 2007. It has been recognized as a public health issue with approximately 2.5 million envenomation accidents worldwide, leading to 125,000 deaths and many victims with serious permanent sequelae [1]. Although the physiopathology of these accidents is a complex event, it is known that inflammatory mediators play an important role in the

**Funding:** The author(s) received financial support from Comissão de Aperfeiçoamento de Pessoal de Nível Superior (CAPES).

**Competing interests:** The authors have declared that no competing interests exist.

envenomation, dynamics, and this has been demonstrated in diverse studies in experimental models and humans [2–6].

In Brazil, the *Crotalus durissus* species was responsible for 23.264 accidents over a 10-year period (2007–2017), and represent the most common cause of ophidian accident in the Northeast region [7]. These accidents are characterized by local pain and swelling [8–10], with can be associated with neuromuscular block, acute respiratory distress [9,11], coagulation disturbances [11–13] and acute renal failure (ARF) [12,14,15]. Crotoxin, convulxin, gyroxin and crotamine toxins may be associated with most of these clinical manifestations [16–19].

Generally, there is an imbalance in the recruitment of immune response cells after envenomation, including the T *helper* (Th) lymphocyte, that can be subdivided into subpopulations (Th1, Th2 or Th17) with different immune response functions and cytokine profiles [20–23]. The Th1 lymphocytes are characterized by the production of IFN-γ, TNF, IL-2 and IL-12, affecting the activation of phagocytes, recruitment and lymphocyte TCD8+ activation. Th2 profile produces IL-4, IL-5, IL-10 and IL-13, all involved in the activation of eosinophils, mastocytes, and B-lymphocytes, as well as producing class IgE antibodies. The Th17 profile lymphocytes produce IL-17 and IL-22, which appear to be related to acute and neutrophil inflammatory responses [24,25].

Cytokine production in response to envenomation is an area increasingly under study. Evidence suggests that bothropic and crotalic venom induce elevated levels of IL-1β, IL-6, IL-10, TNF and IFN, contributing to a leukocyte influx [2,4,5,26,27]. This compromises different organs, including the lungs, leading to respiratory failure, septic shock and multiple organ and systems failure [2,13,28,29].

Pulmonary alterations coupled with the action of *C. durissus* venom, such as leukocyte recruitment, congestion, hemorrhage, atelectasis and emphysema have been previously reported [3,30,31]. However, the inflammatory mediators that may be involved in pulmonary damage provoked by venom are still not fully understood.

The present study evaluated the systemic inflammatory response and pulmonary lesions induced by *Crotalus durissus cascavella* venom in mice, by quantifying TNF, IL-6, IL-4, IL-10, IFNγ, IL-17A and IL-2 cytokines and evaluating histopathologic alterations in the pulmonary parenchyma.

## Materials and methods

### *Crotalus durissus cascavella* venom from the state of Bahia

*Crotalus durissus cascavella* venom from a Caatinga ecosystem of Bahia, Brazil, was collected individually of specimens kept in the Scientific Breeding of Venomous Animals of the State University of Feira de Santana, geographic coordinates Latitude 12˚ 16'00" S Longitude 38˚ 58'00" W. The bioterium is homologated by the Brazilian Institute of Environment and Renewable Natural Resources (IBAMA), registration of Federal Registration Number 480922 and by SisGen (Sistema Nacional do Patrimônio Genético e do Conhecimento Tradicional Associado (protocol numbers ABC319C). After extraction, the venom was vacuum-dried and stored at -20˚ C until analyses. Venom protein concentration was determined using bovine serum albumin (Sigma, Chemical Company) as the protein standard [32].

### Animals

Male Swiss mice weighing from 18-22g were supplied by Central Rodent Bioterium of Breeding and Experimentation at State University of Feira de Santana and kept in a controlled environment in a 12/12 hour light-dark cycle. Animals had access to an *ad libitum* supply of food and water. Expression of species-specific behaviors were favored by adequate housing.

Throughout the experiment, careful manipulation of the animals was performed only when necessary in a noise free environment. Concerned parties ensured the highest level of comfort possible and animals welfare, minimal suffering and euthanasia with brevity. We have avoided analgesics and anesthetics as they may interfere both in cytokines production and costimulatory signals required for antigen presentation and T lymphocyte differentiation. The research team, under the permanent supervision of the veterinarian, monitored animals throughout the entire experiment. Observations were conducted every hour to evaluate clinical behavior, level of activity, posture, temperament, locomotion, water and food intake. Signs of respiratory distress, pain or abnormal behavior have been evaluated and recorded. Animals were euthanized immediately at protocol predetermined times (1h, 3h, 6h, 12h, 24h and 48 hours) with an overdose with ketamine (100mg/Kg) and xylazine (10mg/kg) with immediate extraction of peritoneal fluid and lung tissue. No animals were found dead during the experiment period and all mice were euthanized at the predetermined times.

## Ethics statement

The study was conducted in accordance with the ethical principles for animal research adopted by the Brazilian Society of Animal Science and the National Brazilian Legislation n˚.11.794/08 and was approved by the State University of Feira de Santana Animal Ethics Committee (CEUA-UEFS—protocol number 006/2018). The research team received obligatory training in animal welfare, handling and sacrifice technics.

## Inflammatory response in mice induced by *Crotalus durissus cascavella* venom

The experiments were performed a total of 72 mice subdivided into control group (n = 5) and experimental group (n = 7) for each determined time. The experimental group received the venom challenge dose of 50μg/kg [3], by the intraperitoneal (i.p.) route, diluted in 500μL of sterile saline solution (NaCl 0.9%) [3]. The control group received 500μL of sterile saline solution (NaCl 0.9%). Both groups were monitored during the entire experiment (1, 3, 6, 12, 24 and 48 hours). The animals were euthanized with an overdose with ketamine (100mg/Kg) and xylazine (10mg/kg) to obtain the peritoneal fluid and lung.

## Obtaining the peritoneal fluid

In both groups, 2mL of phosphate saline solution (Phosphate Buffered Saline—PBS) was injected into the peritoneal cavity to obtain the peritoneal fluid. Then, the animals had their abdomens massaged to wash the entire cavity. The peritoneal fluid was extracted using a syringe, centrifuged at 3000rpm at 4˚C, for 20 minutes [33] and the supernatant separated and stored at -20˚C for subsequent cytokine dosages.

## Lung tissue disintegration and histopathologic analysis

The inferior lobe of the right lung was removed and stored in 1mL of PBS solution, then, macerated and centrifuged at 3000 rpm at 4˚C, for 20 minutes [33]. Supernatant was separated and stored at 20˚C for subsequent cytokine dosage. The left lung and the superior lobe of the right lung were initially fixed in 10% buffered formalin for a maximum of 48 hours. Fragments underwent ethanol and xylene dehydration and were diaphanized and cut to a width of 4μm and subsequently stained in a hematoxylin-eosin solution (HE) and Masson's trichrome. The cuts were examined, and images captured by Olympus BX 51 microscope with a coupled digital camera (DP25) and digitalized on cellSens software. Morphometric analysis was performed

on five randomly selected microscopic fields on lung parenchymal slides in Swiss mice at different exposure times (1h, 3h, 6h, 12h, 24h, 48h). The number of inflammatory cells was counted in $mm^2$ with a 20x eyepiece and a 20x objective, in an area of 60 $mm^2$. Collagen deposition was quantified by counting the areas marked by Masson's trichrome in $mm^2$ in five randomly selected microscopic fields on the lung parenchyma slides at 6h, 12h, 24h exposure times. All data obtained were analyzed using ImageJ software (USA). Statistical results were evaluated by the nonparametric Mann-Whitney Test with significance of $p < 0.05$. The graphs were generated by GraphPad Prism 5.0 (GraphPad, San Diego, CA, USA).

### Cytokine measurements using Cytometric Bead Array (CBA)

Cytokine concentrations in the peritoneal fluid supernatant and macerated lungs were determined using a BD™ CBA Mouse Th1/Th2/Th17 Cytokine Kit (BD Biosciences, USA) with a FACSCalibur flow cytometer (San Francisco, BD Biosciences). Cytometric Bead Array analysis allowed the simultaneous detection of cytokines, TNF, IL-6, IL-4, IL-10, IFNγ, IL-17A and IL-2, and was performed in accordance with the manufacturer´s instructions. In brief, the Th1/Th2/Th17 cytokine standards were prepared using a vial of lyophilized Mouse and Assay Diluent using the serial dilutions technique. Capture beads were added into each tube containing samples and standards and then incubated for 2 hours at room temperature in the absence of light. The samples were washed with 1 mL buffer at 200g for 5 minutes and resuspended in 150μL wash buffer. Data acquisition was performed using FCAP Array v2.0 software (Soft Flow, Hungary).

### Statistical analysis

The data distribution was evaluated using the Kolmogorov-Smirnov test. To compare the time intervals evaluated, normal distribution data were analyzed using *one-way analysis of variance* (ANOVA) and Tukey´s test. The Kruskall-Wallis test followed by Dunn's Multiple Comparison Test was used in case of skewed distribution. The T student test was employed to compare experiment and control groups, and non-parametric distributions through the Mann-Whitney test. Results were expressed as mean (standard deviation—SD) and median (interquartile range) and values of $p < 0.05$ were considered to be statistically significant. Statistical analysis was performed using GraphPad Prism 5.0 (GraphPad, San Diego, CA, EUA).

## Results

### Clinical manifestations induced by *Crotalus durissus cascavella* venom in mice

Experimental animals presented agitated behavior, pruritus and wound licking. One hour after inoculation, subjects showed patterns of nesting behavior, prostration, progressive lethargy and tremors (Fig 1A, 1B and 1C). Peak respiratory discomfort occurs between 6 and 48 hours, characterized by tachypnea with intense abdominal contractions and thoracic effort (Fig 1D and 1E). Animals in the control group showed no clinical deterioration throughout the experimental period.

### Determining Th1/Th2/Th17 cytokine levels in the peritoneal fluid and macerated lung

Analysis of the peritoneal fluid and macerated lung indicated that *Crotalus durissus cascavella* venom induced varied levels of Th1/Th2/Th17 cytokines and regulatory response at all intervals analyzed. Significant differences between the experimental and control groups were found

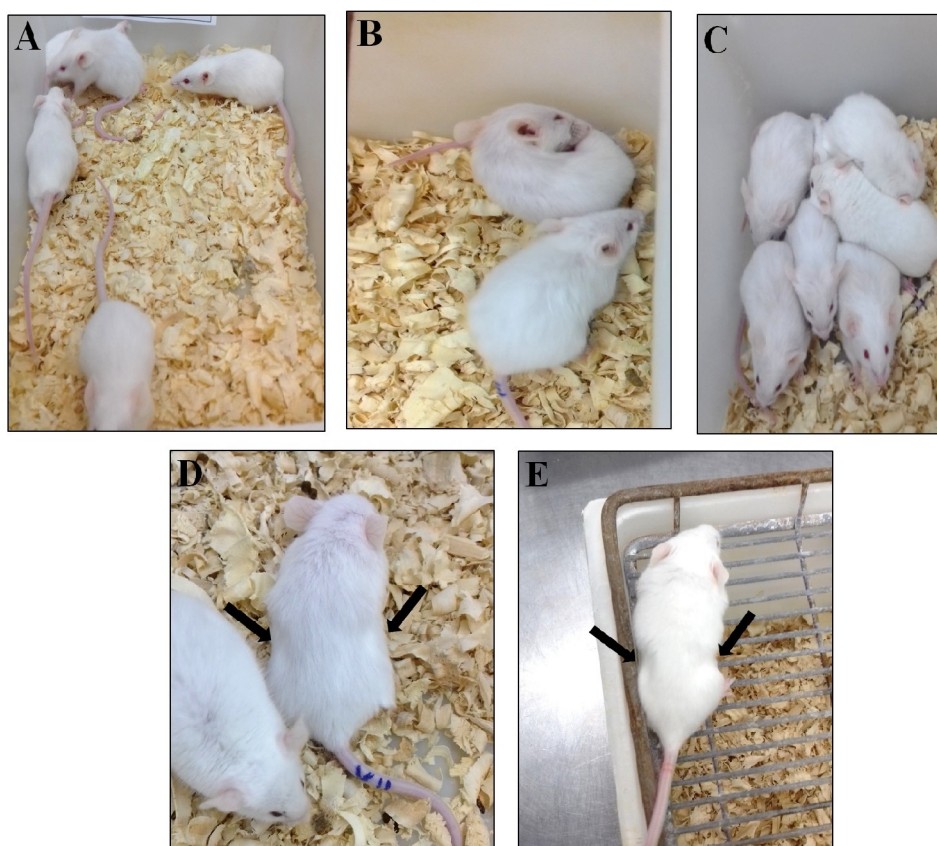

**Fig 1. Mice and their clinical manifestations.** Control group was inoculated (i.p.) with 500µL saline solution (A). Experimental group inoculated (i.p.) with 50µg/kg *Crotalus durissus cascavella* venom diluted in 500µL saline solution (B-D): Initial clinical symptoms observed were pruritus, wound licking (B) and nesting behavior (C); after three hours the animals presented lethargy and respiratory distress (D); 24 hours after inoculation, there were intense abdominal contractions and thoracic effort (E).

in TNF (Fig 2A), IL-6 (Fig 2B) and IL-4 (Fig 2C) concentrations in the peritoneal fluid. The experimental group presented significantly higher TNF (#p<0.05) and IL-6 (#p<0.05) 1 hour after inoculation. IL-6 levels of the experimental group continued to be higher than the control group at the 3-hour temporal window (#p<0.05). There was an increase in IL-4 concentration at the 24-hour time (#p<0.05).

Despite not statistically significant, IL-10 (Fig 2D) levels were detected after 12 hours, peaking at 48 hours. IL-17A (Fig 2E) and IFN-γ (Fig 2F) had increased levels at 24 hours with subsequent decline 48 hours after exposure, while IL-2 (Fig 2G) levels remained constant.

Analyzing the kinetics of cytokine production in the peritoneal fluid of the experimental group, TNF reaches highest levels over the first hour with subsequent gradual decline (*p<0.05 e **p<0.01). IL-6 also presented a statistically significant difference in the experimental group (*p<0.05 e **p<0.01), with a peak in the first hour and a subsequent progressive decline over the other time windows. IL-4 production remained similar in the first time windows (1, 3 and 6h), dropping off at 12 hours, with elevated late levels at 24 hours (#p<0.05). Levels of IFNγ, IL-17A and IL-2 have not presented a statistically significant difference.

Among cytokines in the macerated lungs (Fig 3), both IL-4, at 3 hour and IL-10, at the 3 and 12 hour times, demonstrated significant difference between the experimental and the control groups (#p<0.05). The remaining analyzed cytokines have not showed significant

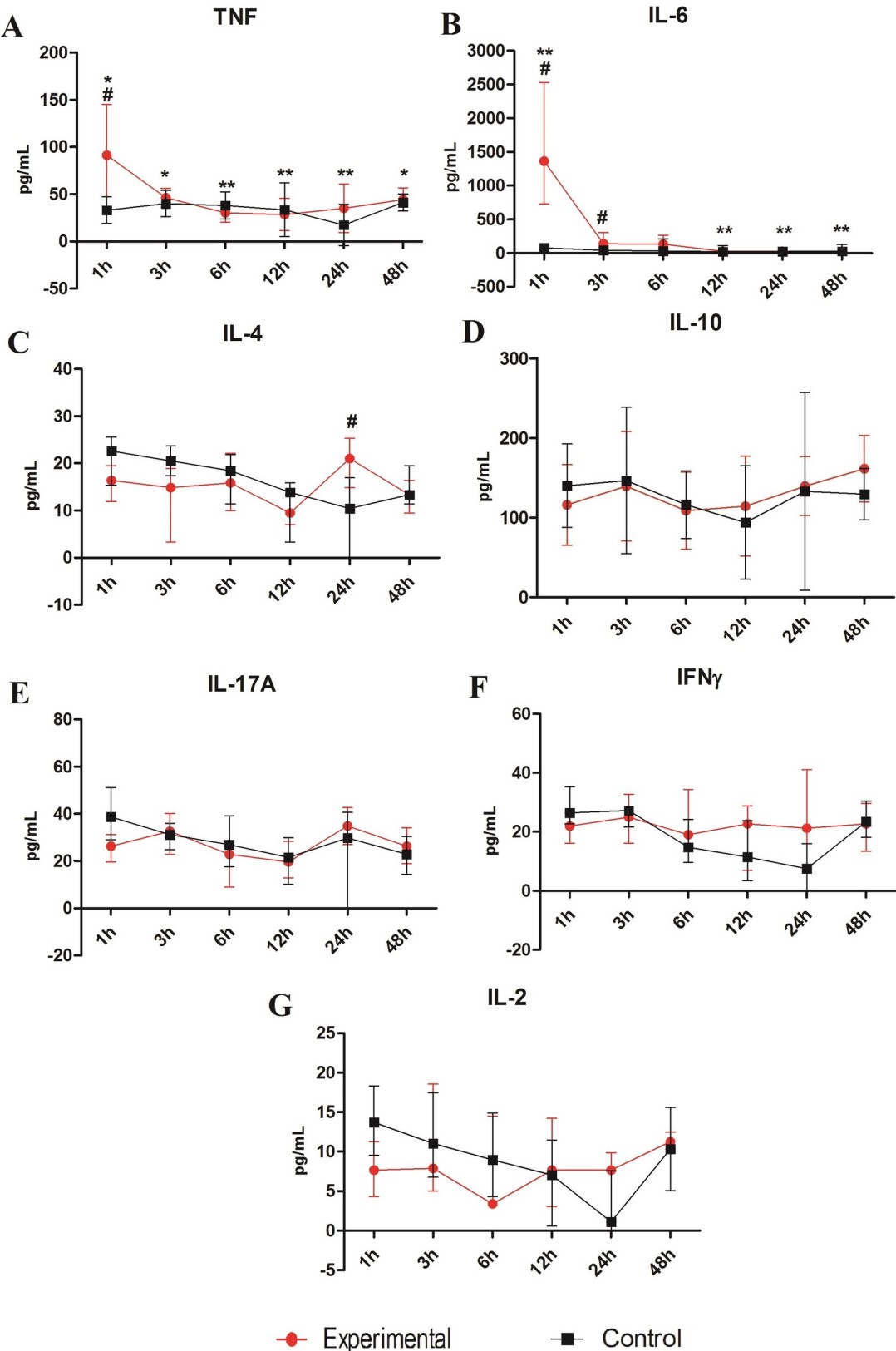

**Fig 2.** Cytokine levels TNF (A), IL-6 (B), IL-10 (C), IL-4 (D), IL-17A (E), IFN-γ (F) and IL-2 (G) measured in peritoneal fluid of the Swiss mice at different time 1, 3, 6, 12, 24 and 48 hours. The mice were inoculated via i.p with 50μg/kg *Crotalus durissus cascavella* venom diluted in 500μL saline solution. The control group animals were inoculated with 500μL sterile saline solution.

Each point represents the mean–SD (A and C) and median—IQR (B, D, E, F and G) of animals per group. *p< 0.05 and **p<0.01 in relation to the venom treatment times and #p<0.05 when compared to the control group.

differences. During follow up period, only IFN-γ presented a statistically significant profile between the 3 and 48 hour time windows (*p<0.05), peaking at 3 hours and declining to its lowest level at 48 hours.

## Histopathology of lung damage induced by *Crotalus durissus cascavella* venom

There was no physiopathological alteration of the pulmonary parenchyma in control group (Fig 4A). Nevertheless, *Crotalus durissus cascavella* venom induced early recruitment of

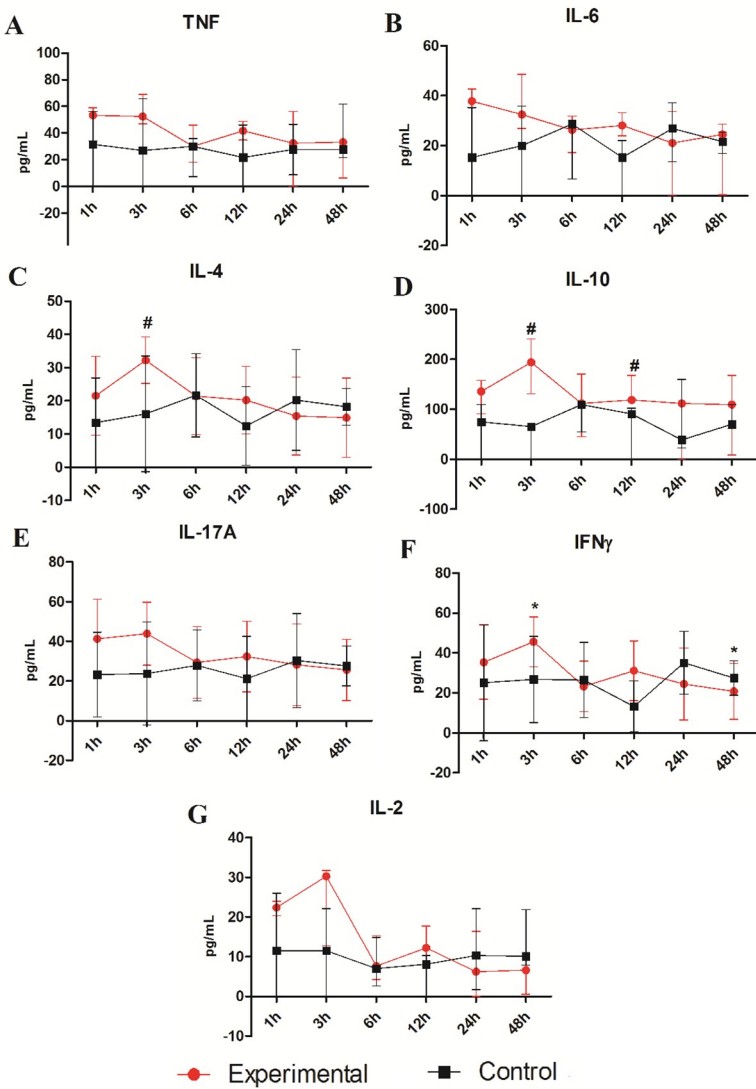

**Fig 3.** Cytokine levels TNF (A), IL-6 (B), IL-4 (C), IL-10 (D), IL-17A (E), IFN-γ (F) and IL-2 (G) measured in the macerated lungs of the Swiss mice at different time 1, 3, 6, 12, 24 and 48 hours. The mice were inoculated via i.p with 50μg/kg *Crotalus durissus cascavella* venom diluted in 500μL saline solution. The control group animals were inoculated with 500μL sterile saline solution. Each point represents the mean–SD (C, E and F) and median–IQR (A, B, D and G) of animals per group. *p< 0.05 in relation to the venom treatment times and #p<0.05 when compared to the control group.

inflammatory cells, thickening of alveolar septa and vascular congestion in the initial phase (1 hour) (Fig 4B). At 3 hours, there was a significant increase of peribronchial inflammatory infiltrates, emphysema and focal atelectasis indicative of severe pulmonary inflammation (Fig 4C).

At six hours, a decrease in the activity of inflammatory response is observed, despite an increase in vascular congestion, alveolar septa thickening and emphysematous areas (Fig 4D). Between 12 and 48 hours, there were perceptible inflammatory infiltrates in the pulmonary parenchyma, hemorrhagic focuses, vascular congestion and bronchial muscle distension (Fig 4E).

An intense chronic inflammatory infiltrate was observed between 24 and 48 hours, consisting mainly of neutrophils and eosinophils, compressed bronchiole, atelectasis and bronchial muscle distension (Fig 4F e 4G). The morphometric quantification of polymorphonuclear cells in inflammatory infiltrates in pulmonary parenchyma showed significant difference between the times of 1 and 3 hours (p = 0,00159); 1 and 6 hours (p = 0,00317); the times 3, 6, 12 and 24 hours (p = 0,0079); 3 and 48 hours (p = 0,00317), 6 and 12 hours (p = 0,0079), 6 and 24 hours (p = 0,0159); 6 and 48 hours (p = 0,0079), 12 and 24 and between 12 and 48 hours (p = 0,0079). No statistical difference was showed between concentrations at 24 and 48 hours (Fig 5). Masson's Trichrome staining revealed matrix changes with collagen deposition at 6, 12 and 24 hours (Fig 6), although without statistically significant difference, as confirmed by quantification (Fig 7).

## Discussion

Physiopathogenic mechanisms that lead to pulmonary damage induced by *Crotalus durissus* venom are still not fully understood, especially inflammatory response and cytokine production [2,28,30,31]. The present study observed a systemic inflammatory response induced by *Crotalus durissus cascavella* venom associated with aggressive acute pulmonary injury characterized by peribronchial inflammatory infiltrate and altered vascular permeability with patchy hemorrhagic foci.

In response to acute antigenic stimulus induced by toxins, a modulated Th1, Th2 or Th17 immune response occurs [24,25]. The systemic action of toxins triggers an inflammatory response and enhance production of several immunological mediators that activates recruitment, proliferation and differentiation of leukocytes [29,34].

An early increase of IL-6 and TNF levels in the peritoneal fluid and macerated lungs suggests that pro-inflammatory cytokines play a key role in acute inflammation [29]. Previous studies have reported a similar inflammatory profile in the first hours after inoculation of *C. d. terrificus* venom with subsequent late immunomodulation. These characteristics can be related to crotoxin activity in inhibiting neutrophil chemotaxis and modulating adaptive immune response [17,21,35–37].

Snakebite envenoming induces high levels of IL-6, TNF and IL-1β cytokines and may provoke fever, lethargic, vasodilation and a variety of other symptoms resulting from edema induction, T and B cell activation, and leukocyte recruitment [2,38,39]. It is plausible that high levels of these cytokines are directly related to aggressive clinical manifestations, such as prostration, lethargy and huddling.

Increased cytokines levels during the acute stage have been described by several authors in accidents involving venomous animals [4,33,40–42]. Studies involving scorpion venom demonstrated a positive physiopathological correlation between cytokine concentrations and severity of clinical manifestations [43–46]

Systemic toxicity is markedly affected by cytokines release. Increase IL-10 levels in peritoneal fluid at 12 hour and at 3h and 12h in macerated lungs may play an important role in

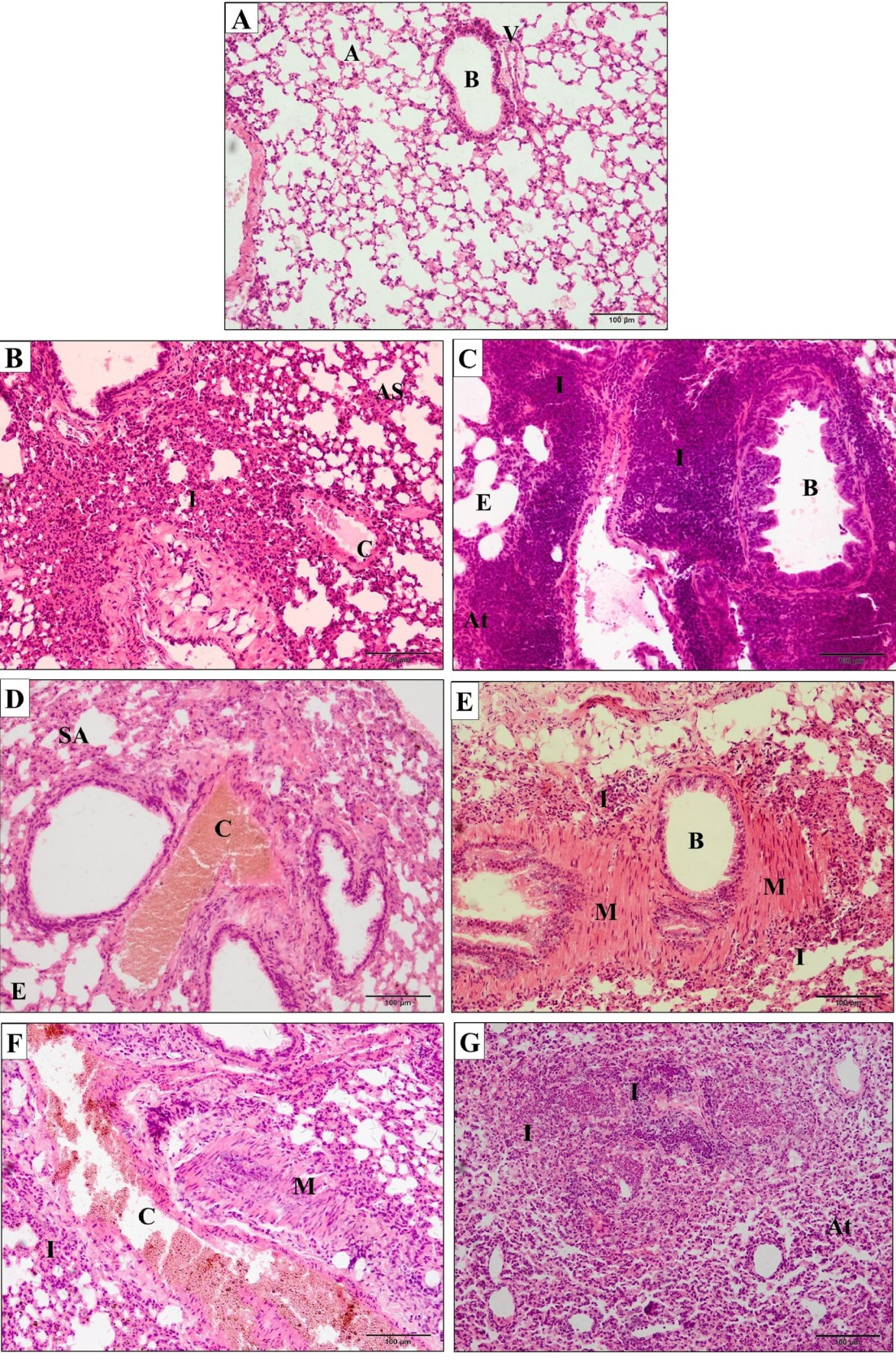

**Fig 4. Photomicrographs of pulmonary parenchyma of mice.** (A) Control group inoculated with 500μL sterile saline solution presented preserved pulmonary architecture (A–alveolus; B–bronchial; V–vessels). (B–G) Experimental groups inoculated with 50μg/kg *Crotalus durissus cascavella* venom at different observation time: (B) 1h; (C) 3h; (D) 6h; (E) 12h; (F) 24h; (G) 48h. (I–Inflammatory infiltrates; AS–alveolar septum thickening; C–vascular congestion; E–emphysema; At–atelectasis; M—bronchial muscle distension).

immune response modulation, favoring a shift from a Th1 towards a Th2 cytokine. The increase in IL-4 in macerated lungs at 3h and at 24h in peritoneal fluid strengthened this hypothesis. Similar data were observed with *C. d. terrificus* venom [29]. It was also shown that *Tityus serrulatus* scorpion venom induced an increase in IL-10 and a decrease in IL-6 and TNF [46].

Cardoso and Sampaio et al. [21,47] also reported that inhibitory effects induced by crotoxin on chemotaxis and macrophage phagocytosis may increase IL-10. Furthermore, phospholipase A$_2$ (PLA$_2$), a subunit of crotoxin, could trigger IL-4 production through mastocyte activation [38].

Anaphylaxis is a severe acute allergic manifestation with potentially fatal clinical repercussions, usually prompted by IgE mediated hypersensitivity. The levels of IL-4 become particularly elevated and induce an increase susceptibility to vasoactive mediators [48]. Previous studies observed anaphylactic response to snake venom characterized by rising serum IgE levels. These observations suggest that increase levels of IL-4 and IL-6 may be related to the anaphylactic response in the experimental group. Further research is required to provide evidence of this anaphylactic pathway due to their potential implications for the clinical management of patients bitten by *Crotalus durissus cascavella*.

The low levels of IL-2 found in the macerated lungs and peritoneal fluid indicated an inhibitory effect on lymphoproliferation, as observed in studies *in vitro* with *C. d. terrificus* and *C. d.*

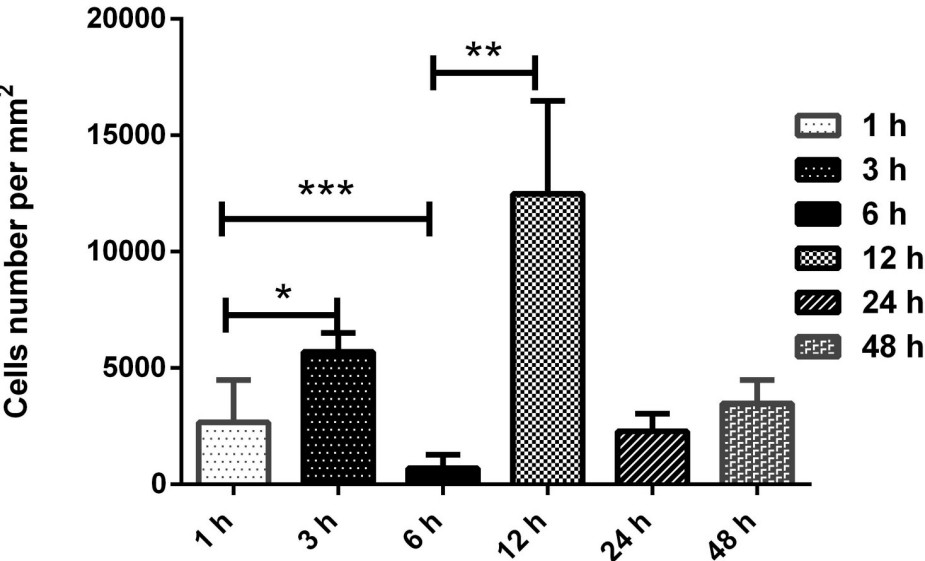

**Fig 5. Morphometric analysis of the frequency of polymorphonuclear cells in inflammatory infiltrates in pulmonary parenchyma of Swiss mice.** Experimental groups inoculated with 50μg/kg *Crotalus durissus cascavella* venom at different observation time showed significant differences with p = 0,00159 (1 and 3 hours); p = 0,00317 (1 and 6 hours); p = 0,0079 (3, 6, 12 and 24 hours); p = 0,00317 (3 and 48 hours); p = 0,0079 (6 and 12 horas); p = 0,0159 (6 and 24 hours); p = 0,0079 (6 and 48 hours); p = 0,0079 (12 and 24/ 12 and 48 hours). No statistical difference was showed between concentrations at 24 and 48 hours. Statistical analysis was performed by the bilateral Mann—Whitney Test. *p<0,05 were considered statistically significant.

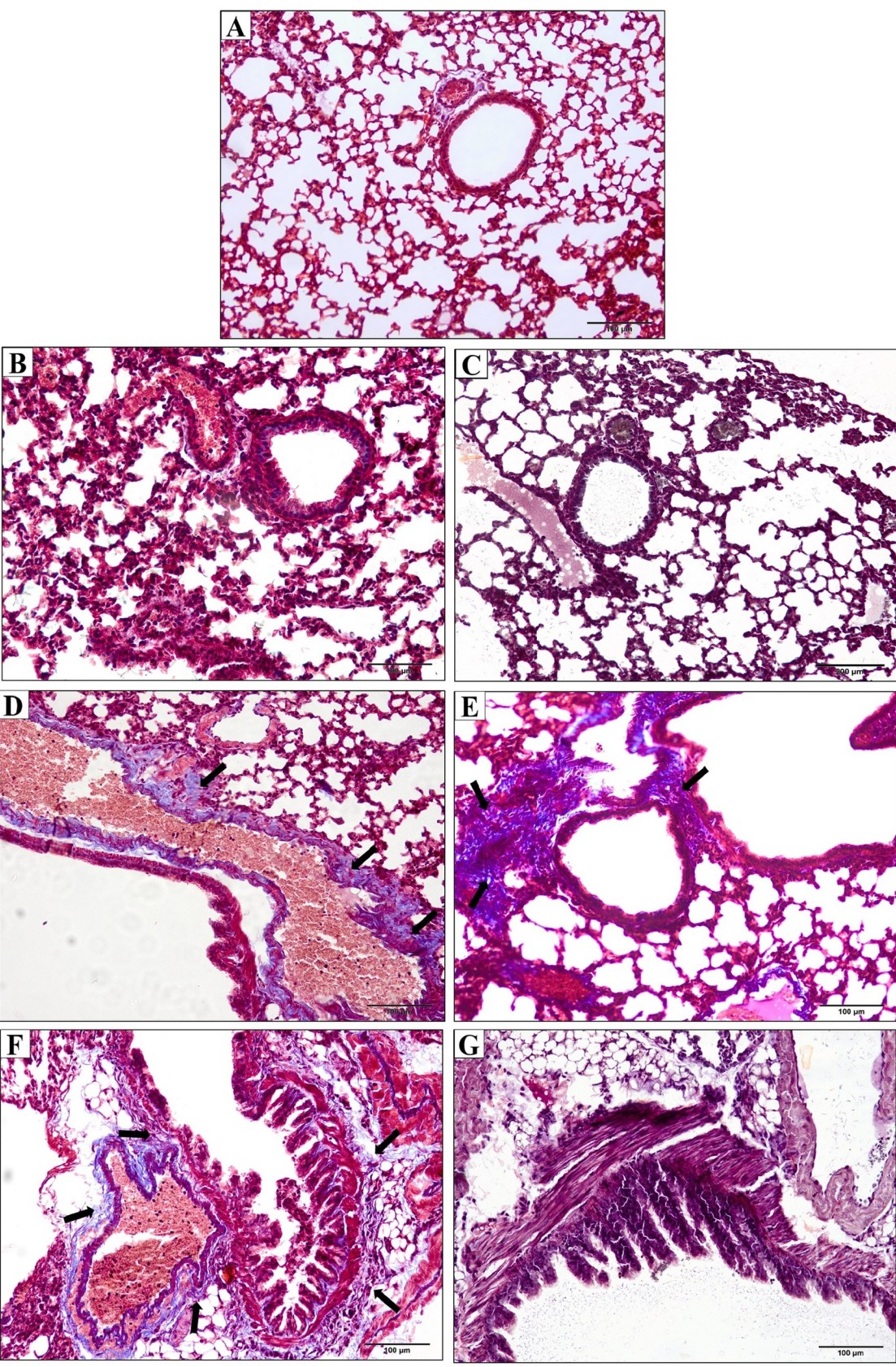

**Fig 6. Photomicrographs of pulmonary parenchyma of mice stained with Masson's trichrome.** (A) Control group inoculated with **500μL** sterile saline. (B-G). Experimental groups inoculated with 50μg/kg *Crotalus durissus cascavella* venom at different observation time: (B) 1h; (C) 3h; (D) 6h; (E) 12h; (F) 24h; (G) 48h. The arrows indicate collagen deposits (stained in blue) in the peribronchial and perivascular regions (Masson's trichrome staining).

*collilineatus* venom, and with *C. d. terrificus* crotoxin [17,49]. However, short half-life and low circulation of IL-2 [50] may reduce detection levels of this cytokine.

Decline of IFNγ in macerated lungs and the absence of significant levels of IL-17 may be related to IL-4 production. This cytokine plays a pivotal role decreasing the production of Th1 and Th17 cells. However, the increase in IFNγ in the first hours (3h) in macerated lungs, despite the presence of IL-4 and IL-10, may lead to systemic disorders such disseminated vascular coagulopathy [3]. Furthermore, metalloproteinases can activate prothrombin and coagulation factor X, which inhibit platelet aggregation, and lead to apoptotic activity and hemostatic changes [51].

Despite its severity, few studies have examined pulmonary alterations in snakebite accidents [28,52–54]. In crotalic accidents, crotoxin blocks the neuromuscular transmission that contributes to the development of paralysis, muscular respiratory insufficiency and acute respiratory distress [11,28]. The respiratory abnormalities related in this study are similar to the severe cases of respiratory paralysis reported in accidents caused by snakes in the Elapidae family [55,56], and *Vipera palaestine* [57].

Hemorrhage focuses, congestion, atelectasis and emphysema in the pulmonary parenchyma within 3 hours of exposure demonstrate an early physiopathological pathway of pulmonary damage and impaired respiratory mechanics, leading to acute lung injury (ALI). Similar alterations were observed with *Crotalus durissus cascavella* and *Tityus* e *Androctonus* venom [31,58–61]. Respiratory damage have also been attributed to *Bothrops jararaca* venom, either

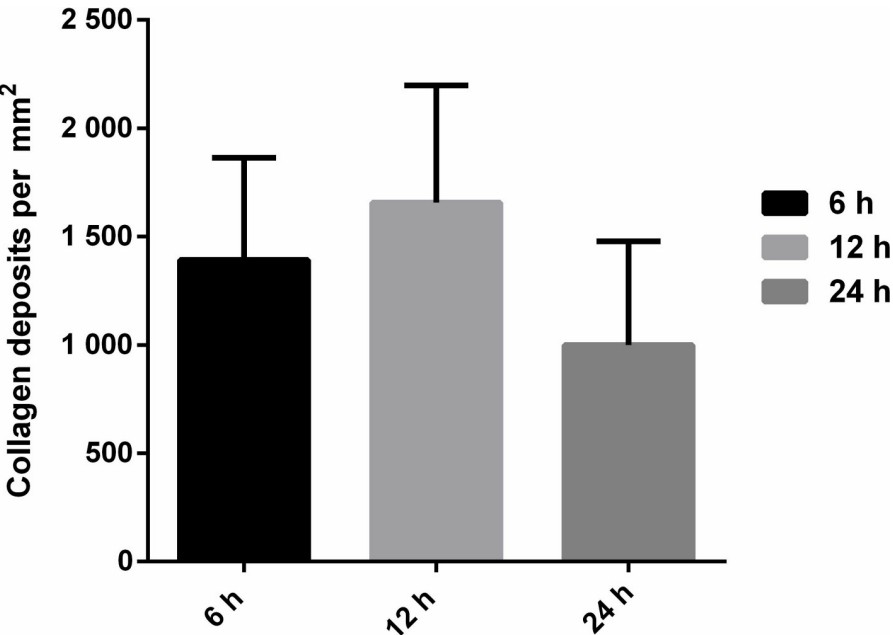

**Fig 7. Quantification of collagen deposits in Swiss mice inoculated with 50μg/kg *Crotalus durissus cascavella* venom at different observation time.** Evaluation of collagen deposits by Masson's trichrome staining revealed no significant differences at any of the correlated points. Statistical analysis performed by the bilateral Mann-Whitney test. p<0,05 was considered statistically significant.

by hemorrhage due metalloproteinases or by the action of PLA2 which can lead to pulmonary inflammation [62]. These data emphasize that snakebite victims presenting respiratory alterations require intensive monitoring.

Massive PMN cells proliferation in the pulmonary parenchyma, present up to the 48 hour time window, is probably linked to an increase in IL-17 levels [30,31]. Collagen deposition in the pulmonary parenchyma observed at 6, 12 and 24 hours can be related to mechanisms of repair and remodeling following an aggressive inflammatory response [63,64].

In summary, these findings advance our understanding of the pathophysiologic events and acute lung injury (ALI) induced by *Crotalus durissus cascavella* venom. High levels of acute phase cytokines detected in the peritoneal fluid and macerated lungs induced a systemic inflammatory response. There is a positive correlation between severity of clinical manifestations and pro-inflammatory cytokines levels (IL-6, TNF). Nevertheless, enhanced IL-4 production indicates a dynamic switch from Th1 to Th2 response and may be related to anaphylaxis triggered by venom components. Public policies should to be designed to prevent and improve patient outcomes in snake accidents.

## Acknowledgments

The authors are grateful to Comissão de Aperfeiçoamento de Pessoal de Nível Superior (CAPES), Laboratory of Venomous Animals and Herpetology team, especially M. Nolasco, I. Cabral, D. Andrade and E. Dias for their assistance in the experimental work and Dr. L. Gusmão of the State University of Feira de Santana—Mycology Laboratory for making the microscope available for image acquisition.

## Author Contributions

**Conceptualization:** Elen Azevedo, Ilka Biondi, Soraya Castro Trindade.

**Data curation:** Elen Azevedo.

**Formal analysis:** Elen Azevedo, Geraldo Pedral Sampaio, Marcos Lázaro da Silva Guerreiro, Soraya Castro Trindade.

**Funding acquisition:** Ricardo Gassmann Figueiredo, Ilka Biondi, Soraya Castro Trindade.

**Investigation:** Elen Azevedo, Geraldo Pedral Sampaio, Rebeca Pereira Bulhosa Santos.

**Methodology:** Elen Azevedo, Marcos Lázaro da Silva Guerreiro, Ilka Biondi, Soraya Castro Trindade.

**Project administration:** Elen Azevedo, Ilka Biondi, Soraya Castro Trindade.

**Resources:** Elen Azevedo, Roberto Vieira Pinto, Tarsila de Carvalho Freitas Ramos, Marcos Lázaro da Silva Guerreiro, Ilka Biondi, Soraya Castro Trindade.

**Software:** Elen Azevedo.

**Supervision:** Ilka Biondi, Soraya Castro Trindade.

**Validation:** Ricardo Gassmann Figueiredo, Ilka Biondi, Soraya Castro Trindade.

**Writing – original draft:** Elen Azevedo, Ilka Biondi, Soraya Castro Trindade.

**Writing – review & editing:** Elen Azevedo, Ricardo Gassmann Figueiredo, Roberto Vieira Pinto, Ilka Biondi, Soraya Castro Trindade.

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
