## [Decision Letter · Decision Letter 0]

26 Nov 2019

PONE-D-19-26162

EVALUATION OF SYSTEMIC INFLAMMATORY RESPONSE AND LUNG INJURY INDUCED BY Crotalus durissus VENOM

PLOS ONE

Dear Figueiredo,

Thank you for submitting your manuscript to PLOS ONE. After careful consideration, we feel that it has merit but does not fully meet PLOS ONE’s publication criteria as it currently stands. Therefore, we invite you to submit a revised version of the manuscript that addresses the points raised during the review process.

More specifically, as reviewer #2 commented, cell characterization of blood samples is important to consider a systemic response. Also, morphological quantification of histological slides must be performed. Both reviewers have some issues about the dose used as, by the criterion chosen, should be lethal for part of the experimental group. This should be clearly explained. Point-by-point answers to the reviewers are recommended. 

We would appreciate receiving your revised manuscript by Jan 10 2020 11:59PM. To enhance the reproducibility of your results, we recommend that if applicable you deposit your laboratory protocols in protocols.io, where a protocol can be assigned its own identifier (DOI) such that it can be cited independently in the future. For instructions see: http://journals.plos.org/plosone/s/submission-guidelines#loc-laboratory-protocols

We look forward to receiving your revised manuscript.

Kind regards,

Luis Eduardo M Quintas, Ph.D.

Academic Editor

PLOS ONE

Journal Requirements:

2. At this time, we request that you please report additional details in your Methods section regarding animal care, as per our editorial guidelines. Specifically, please describe the following points:

(1) The criteria used to assess animal health and well-being during the 48 hour time course of Crotalus durissus venom administration.

(2) The dosage of the venom applied to the mice in relation to the weight of the animals (e.g. ug/kg)

Thank you for your attention to these requests.

3. Please ensure that you include a title page within your main document. You should list all authors and all affiliations as per our author instructions and clearly indicate the corresponding author..

Reviewers' comments:

Reviewer's Responses to Questions

**Comments to the Author**

1. Is the manuscript technically sound, and do the data support the conclusions?

Reviewer #1: Yes

Reviewer #2: Partly

2. Has the statistical analysis been performed appropriately and rigorously? 

Reviewer #1: Yes

Reviewer #2: Yes

3. Have the authors made all data underlying the findings in their manuscript fully available?

Reviewer #1: No

Reviewer #2: Yes

4. Is the manuscript presented in an intelligible fashion and written in standard English?

Reviewer #1: Yes

Reviewer #2: Yes

5. Review Comments to the Author

Reviewer #1: The authors have produced some interesting data that contributes to understanding the neglected health problem of snake envenoming. In order to better place this contribution a few items could be addressed:

1) Information on which C. durissus subespecies provided the venom used in this research could be important, once C. durissus terrificus seems to be the most common cause of accidents by this genus as showed by reference 8 cited by the authors and would be nice to be provided.

2) The authors described the use of 1 LD 50 in this study (lines 113-5), that the "Venom dose of 1.0 µg is the lethal dose (1LD50%) for the Crotalus durissus that inhabit the State of Bahia. This dose has been previously determined in collaboration with Butantan National Institute" (lines 86-7 without reference) and also that "no animals were found dead during the experiment period" (lines 102-3). Since by definition LD50 is the amount of substance that causes death of 50% of individuals on a test group, how come no test individual died? Adding the reference to previously determined LD 50 (lines 86-7) and/or revising this information on the final manuscript might clear this issue. Also, if less than LD50 was intentionally chosen to spare animal use in this study it should be specified in the methods.

3) The use of parametric and non-parametric statistical tests may create some interference in understanding the results. Adding information on graphs or legends of figures 2 and 3 indicating each test was used where and using MEAN±SEM data should clear this issue.

4) The correlation of the data observed in this paper with scorpions venoms and other snake venoms with neurotoxins is well explored by the authors as partially responsible for the respiratory distress observed. However there is also some literature on other snake venom genus and species without pronounced neurotoxic effects venoms inducing lung damage with inflammatory response as showed for example by Silveira, et al 2004 for Bothrops jararaca. This should be considered in the discussion in the attempt to exclude the neuromuscular block effect bias from the inflammatory induced lung damage.

Besides the above highlighted points some attention in form on abbreviations, figure number on text and a few minor misspelling corrections will be needed.

Reviewer #2: The study " EVALUATION OF SYSTEMIC INFLAMMATORY RESPONSE AND LUNG INJURY INDUCED BY Crotalus durissus VENOM" has a descriptive approach about a systemic inflammatory response induced by Crotalus durissus venom associated with aggressive and acute pulmonary injury.

Some minor corrections can be done.

On pdf page 11, line 137: use " with a coupled digital camera " instead of "with an coupled digital camera".

On page 13, line 204: revise Figure 09 citation, this is out of the sequence.

Figure 5 legend need to be corrected: There is text written in Portuguese “Coloração TM, objetiva”

Some Considerations:

1) It is critical to defend systemic inflammatory response without investigate blood sample. Why the authors choose peritoneal lavage instead of serum?

2) The authors consider that the dose administered was lethal (see page 10, lines 113 and 114). Data shown in figure 2 and 3 did not demonstrate a severe and lasting inflammation in both sample used, especially in the lungs, once 3 hours after inoculation was the last time-point with increased pro-inflammatory markers. To consider a lethal dose, authors should provide survival curve.

3) An important deposit of connective tissue is observed around airways. I recommend some morphological quantification to validate that.

6. PLOS authors have the option to publish the peer review history of their article (what does this mean?). If published, this will include your full peer review and any attached files.

Reviewer #1: No

Reviewer #2: Yes: Manuella Lanzetti

---

## [Author Response · Author response to Decision Letter 0]

12 Jan 2020

Please find attached here the revised version of our manuscript entitled "Evaluation of Systemic Inflammatory Response and Lung Injury Induced by Crotalus durissus cascavella Venom ", resubmitted for your consideration for publication in Plos One. You will find our point-by-point responses to the editorial revisions below. All changes in the revised text have been highlighted. 

Please feel free to contact us with any questions or concerns, and we eagerly await your response.

Journal Requirements:

R-Additional requirements have been provided.

2. At this time, we request that you please report additional details in your Methods section regarding animal care, as per our editorial guidelines. Specifically, please describe the following points:

(1) The criteria used to assess animal health and well-being during the 48-hour time course of Crotalus durissus venom administration.

R - The research team, under the permanent supervision of the veterinarian, monitored animals throughout the entire experiment. Observations were conducted every hour to evaluate clinical behavior, level of activity, posture, temperament, locomotion, water and food intake. Signs of respiratory distress, pain or abnormal behavior have been evaluated and recorded (photograhs and videos). Throughout the experiment, careful manipulation of the animals was performed only when absolutely necessary in a noise free environment, avoiding discomfort as much as possible, and thus, preventing disturbances in both the macro and the microenvironment. Animals were euthanized immediately at protocol preterminated times (1h, 3h, 6h, 12h, 24h and 48 hours) with an overdose with ketamine (100mg/Kg) and xylazine (10mg/kg) with immediate extraction of peritoneal fluid and lung tissue. Prior to the experiments, the team was participated of lectures on animal welfare, with parallel handling and euthanasia training. The management of the animals was realized in accordance with the ethical regulation in animal research (Brazilian Society of Animal Science and the National Brazilian Legislation n °.11.794/08). 

There was concern about achieving the highest possible level of comfort, minimize animals suffering and improve welfare, as well as euthanasia with brevity. We have avoided analgesics and anesthetics as they may interfere both in cytokines production and costimulatory signals required for antigen presentation and T lymphocyte differentiation (SHEERAN, HALL, 1997; KELBEL, WEISS, 2001; CRUZ et al., 2017)

(2) The dosage of the venom applied to the mice in relation to the weight of the animals (e.g. ug/kg)

R - Was applied 1 μg of the venom in each animal weighting 20g, which was equivalent to 50 μg/kg.

3. Please ensure that you include a title page within your main document. You should list all authors and all affiliations as per our author instructions and clearly indicate the corresponding author.

R- A title page has been included.

Reviewer #1: The authors have produced some interesting data that contributes to understanding the neglected health problem of snake envenoming. In order to better place this contribution a few items could be addressed:

1) Information on which C. durissus subespecies provided the venom used in this research could be important, once C. durissus terrificus seems to be the most common cause of accidents by this genus as showed by reference 8 cited by the authors and would be nice to be provided.

R -We adopted the taxonomic denomination of Crotalus durissus because both traditional taxonomic assessment and cranial geometric morphometry of the Crotalus specimens in the State of Bahia showed some differences from C. durissus cascavella and C. durissus terrificus subspecies that inhabits the northeast and south/southeast regions of Brazil, respectively (Figure A). Our results differed from Wuster et al. 2005. Biondi previously described disseminated intravascular coagulopathy and systemic inflammation as a main pathophysiological effect of the venom (Toxicon 2016). We are currently in collaboration with an Australian center to evaluate DNA fragments of subspecies C. d. cascavella and C. d. terrificus.

All these data are current under a confidential agreement of Intellectual Property signed by the State University of Feira de Santana (UEFS), the Foundation for Research Support of the State of Bahia (FAPESB) and the Vital Brasil Institute (IVB) for the development of a specific antivenom.

Thus, we have revised the manuscript and replaced the nomenclature Crotalus durissus by Crotalus durissus cascavella, as it is actually the most described subspecies in the Northeast region.

2) The authors described the use of 1 LD 50 in this study (lines 113-5), that the "Venom dose of 1.0 µg is the lethal dose (1LD50%) for the Crotalus durissus that inhabit the State of Bahia. This dose has been previously determined in collaboration with Butantan National Institute" (lines 86-7 without reference) and also that "no animals were found dead during the experiment period" (lines 102-3). Since by definition LD50 is the amount of substance that causes death of 50% of individuals on a test group, how come no test individual died? Adding the reference to previously determined LD 50 (lines 86-7) and/or revising this information on the final manuscript might clear this issue. Also, if less than LD50 was intentionally chosen to spare animal use in this study it should be specified in the methods.

We used in our study 1μg/animal dose as the 1 LD50% challenge dose for Crotalus durissus venom extracted from snakes in the State of Bahia. This LD50% curve was tested by Butantan Institute. Once this data was available to our research team, we used Swiss mice provided by the Central Rodent Bioterium of the State University of Feira de Santana (UEFS) to perform several experiments were carried out to evaluate the effects of this venom with different routes of inoculation routes and doses of 0.75μg, 1μg, 1, 5μg, 3μg and 6μg. Animals (18-22g) inoculated with 1 μg of venom died within 5-8 hours (so-called critical phase) or remained alive with severe histopathological tissue damage.

We speculate that the absence of deaths in our study, as well in other previous experiments, may be explained by a higher resistance to cytotoxic effects and adaptative capacity of Swiss Webster mice. Furthermore, animals are not isogenic with variation in the physiological and immunological responses. Therefore, we agree with Reviewers and decided to use venom dose per body weight instead LD50% dose.

3) The use of parametric and non-parametric statistical tests may create some interference in understanding the results. Adding information on graphs or legends of figures 2 and 3 indicating each test was used where and using MEAN±SEM data should clear this issue.

R- We thank the reviewer for the suggestion. The choice of the statistical test was made according to the distribution of data in the various possibilities of comparison found, generating this lack of understanding. Some information has been added in the legends to clarify the data description.

4) The correlation of the data observed in this paper with scorpions venoms and other snake venoms with neurotoxins is well explored by the authors as partially responsible for the respiratory distress observed. However there is also some literature on other snake venom genus and species without pronounced neurotoxic effects venoms inducing lung damage with inflammatory response as showed for example by Silveira, et al 2004 for Bothrops jararaca. This should be considered in the discussion in the attempt to exclude the neuromuscular block effect bias from the inflammatory induced lung damage.

R- We thank the reviewer for the suggestion and have addressed this issue in the discussion section (lines 361-5). 

Besides the above highlighted points some attention in form on abbreviations, figure number on text and a few minor misspelling corrections will be needed.

R- We apologize for the misspelling. A careful review was made throughout the text.

Reviewer #2: The study "EVALUATION OF SYSTEMIC INFLAMMATORY RESPONSE AND LUNG INJURY INDUCED BY Crotalus durissus VENOM" has a descriptive approach about a systemic inflammatory response induced by Crotalus durissus venom associated with aggressive and acute pulmonary injury.

Some minor corrections can be done.

On pdf page 11, line 137: use "with a coupled digital camera" instead of "with an coupled digital camera".

On page 13, line 204: revise Figure 09 citation, this is out of the sequence.

Figure 5 legend need to be corrected: There is text written in Portuguese “Coloração TM, objetiva”

R- We have changed accordingly.

Some Considerations:

1) It is critical to defend systemic inflammatory response without investigate blood sample. Why the authors choose peritoneal lavage instead of serum?

R- Cytokines are molecules of mainly paracrine action, which makes their detection in certain fluids such as serum difficult, regardless of the technique employed. Due to the intense inflammatory process in the peritoneal region, this paracrine activity can be better evaluated. In addition, the inflammatory response observed in the histopathological analysis can provide a more representative interpretation of the response in vivo. Other ongoing histopathological studies are confirming these data, showing inflammatory processes varying from moderate to severe.

2) The authors consider that the dose administered was lethal (see page 10, lines 113 and 114). Data shown in figure 2 and 3 did not demonstrate a severe and lasting inflammation in both sample used, especially in the lungs, once 3 hours after inoculation was the last time-point with increased pro-inflammatory markers. To consider a lethal dose, authors should provide survival curve.

R- The Swiss strain was more resistant to the effects of venom, and we could observe the progress of inflammation. Thus, as well pointed out by this reviewer, no animals died. We changed the text accordingly. Regarding the lack of lasting and severe inflammatory responses, this may be due to the increase of IL-10 and IL-4, that modulate the response by reducing the expression of IL-2, IL-6 and IL-12. This may be one of the mechanisms that promotes resistance to the swiss strain. In addition, the use of PBS to obtain the peritoneal lavage may have led to a lower concentration of cytokines in this fluid, with can explain the differences in comparison with the histopathological and morphometric data.

3) An important deposit of connective tissue is observed around airways. I recommend some morphological quantification to validate that.

R - We thank the reviewer for the recommendation. A morphometric quantification was performed and inserted into the manuscript in the material and method sections (lines 139-148) and results (lines 257-266). A graphic representations of this data was also included (Figs. 5 and 7).

---

## [Decision Letter · Decision Letter 1]

29 Jan 2020

EVALUATION OF SYSTEMIC INFLAMMATORY RESPONSE AND LUNG INJURY INDUCED BY Crotalus durissus cascavella VENOM

PONE-D-19-26162R1

Dear Dr. Figueiredo,

We are pleased to inform you that your manuscript has been judged scientifically suitable for publication and will be formally accepted for publication once it complies with all outstanding technical requirements.

Within one week, you will receive an e-mail containing information on the amendments required prior to publication. For instance, reviewer #2 noted that "control" group is not correct in Fig. 2 (please correct). When all required modifications have been addressed, you will receive a formal acceptance letter and your manuscript will proceed to our production department and be scheduled for publication.

With kind regards,

Luis Eduardo M Quintas, Ph.D.

Academic Editor

PLOS ONE

Additional Editor Comments (optional):

Reviewers' comments:

Reviewer's Responses to Questions

**Comments to the Author**

1. If the authors have adequately addressed your comments raised in a previous round of review and you feel that this manuscript is now acceptable for publication, you may indicate that here to bypass the “Comments to the Author” section, enter your conflict of interest statement in the “Confidential to Editor” section, and submit your "Accept" recommendation.

Reviewer #1: All comments have been addressed

Reviewer #2: All comments have been addressed

2. Is the manuscript technically sound, and do the data support the conclusions?

Reviewer #1: Yes

Reviewer #2: (No Response)

3. Has the statistical analysis been performed appropriately and rigorously? 

Reviewer #1: Yes

Reviewer #2: (No Response)

4. Have the authors made all data underlying the findings in their manuscript fully available?

Reviewer #1: Yes

Reviewer #2: (No Response)

5. Is the manuscript presented in an intelligible fashion and written in standard English?

Reviewer #1: Yes

Reviewer #2: (No Response)

6. Review Comments to the Author

Reviewer #1: (No Response)

Reviewer #2: The authors have adequately addressed the comments. There is a minimal observation to do. At the Figure 2, the mention of "control"group is wrong. Please, review this.

7. PLOS authors have the option to publish the peer review history of their article (what does this mean?). If published, this will include your full peer review and any attached files.

Reviewer #1: No

Reviewer #2: Yes: Manuella Lanzetti

---

## [Editor Report · Acceptance letter]

4 Feb 2020

PONE-D-19-26162R1 

Evaluation of Systemic Inflammatory Response And Lung Injury Induced By *Crotalus durissus cascavella* Venom 

Dear Dr. Figueiredo:

I am pleased to inform you that your manuscript has been deemed suitable for publication in PLOS ONE. Congratulations! Your manuscript is now with our production department. 

With kind regards,

on behalf of

Dr. Luis Eduardo M Quintas 

Academic Editor

PLOS ONE